# Bamboo Lignin Fractions with In Vitro Tyrosinase Inhibition Activity Downregulate Melanogenesis in B16F10 Cells via PKA/CREB Signaling Pathway

**DOI:** 10.3390/ijms23137462

**Published:** 2022-07-05

**Authors:** Moon-Hee Choi, Seung-Hwa Yang, Won-Keun Park, Hyun-Jae Shin

**Affiliations:** 1Department of Beauty and Cosmetics, Graduate School of Industrial Technology Startup, Chosun University, 309 Pilmundaero, Gwangju 61452, Korea; aamoony1222@naver.com; 2Department of Chemical Engineering, Graduate School of Chosun University, Gwangju 61452, Korea; sh556@daum.net; 3Department of Chemical Energy Engineering, Sangmyong University, Seoul 03016, Korea; wkpark@smu.ac.kr

**Keywords:** antioxidant activity, bamboo stem, lignin fractionation, PKA/CREB pathway, steam explosion, tyrosinase inhibition

## Abstract

Cosmetic ingredients originating from natural resources have garnered considerable attention, and the demand for whitening ingredients is increasing, particularly in Asian countries. Lignin is a natural phenolic biopolymer significantly effective as a natural sunscreen, as its ultraviolet protection efficacy ranges from 250 to 400 nm. However, using different types of lignin as cosmetic ingredients is difficult owing to the heterogeneity of lignin and the lack of in vitro and in vivo safety and efficacy data. Thus, steam-exploded lignin (SEL) was prepared from bamboo, fractionated via successive organic solvent extraction, and sequentially fractionated using ethyl acetate, methanol, and acetone to investigate its potential as a natural whitening material. Gel permeation chromatography showed that the molecular weight of acetone-soluble and acetone-insoluble SEL fractions were the lowest and the highest, respectively. Monomer structures of the four lignin fractions were elucidated using ^1^H, ^13^C, and 2D heteronuclear single quantum coherence nuclear magnetic resonance and pyrolysis gas chromatography/mass spectrometry. The antioxidant and tyrosinase inhibition activities of the four fractions were compared. The methanol-soluble SEL fraction (SEL-F2) showed the highest antioxidant activity (except 2,2-diphenyl-1-picrylhydrazyl scavenging activity), and the enzyme inhibition kinetics were confirmed. In this study, the expression pattern of the anti-melanogenic-related proteins by SEL-F2 was confirmed for the first time via the protein kinase A (PKA)/cAMP-response element-binding (CREB) protein signaling pathway in B16F10 melanoma cells. Thus, SEL may serve as a valuable cosmetic whitening ingredient.

## 1. Introduction

Active oxygen, such as reactive oxygen species (ROS), causes photoaging of the skin and produces age spots, spots, and freckles. Moreover, active oxygen elimination correlates with the anti-whitening effect. Recently, effective and safe tyrosinase (TYR) inhibitors for skin-whitening cosmetics have attracted attention in Asian countries [1]. Hydroquinone, a TYR inhibitor also used as a cosmetic whitening agent, is currently banned as a cosmetic ingredient as it is a potential carcinogen. In addition, kojic acid, which has a TYR inhibitory effect and is a representative whitening substance, may be an allergen. Compared to other whitening functional ingredients, arbutin causes less skin irritation and is a safe ingredient, but it may have minor side effects depending on individual skin characteristics [2]. Therefore, natural active components such as plant-based polyphenols and lignin have garnered attention to meet the demand for safe and natural TYR inhibitors. Natural extracts show various effects such as ultraviolet (UV) absorption, anti-inflammatory action, antioxidant action, inhibition of enzymes such as collagenase and TYR, and percutaneous absorption improvement [3]. Recently, side effects such as the production of phenolic synthetic antioxidant butylated hydroxyanisole (BHA) and butylated hydroxytoluene (BHT) have been revealed, and many studies have focused on developing antioxidants from natural sources. Whitening research is broadly classified into two types: the method of inhibiting tyrosinase, the main enzyme of melanin production, and the method of removing the pigmented melanin from the skin. Alpha-melanocyte-stimulating hormone (α-MSH) or proopiomelanocortin (POMC) is a substance that induces melanin formation, and mainly phenolic compounds are known to remove melanin pigmentation [4].

Bamboos belong to the Bambusoideae subfamily, which is a part of the Poaceae family [5]. They are primarily distributed across tropical and subtropical zones, with some in the temperate and frigid zones. Their fast and rhizomatous growth habits are suitable for industrial production because they can be harvested within 4–5 years; furthermore, they can continue growing from the remaining part of their stems in the ground after annual harvests [6]. Bamboo biomass is generally used to manufacture handicrafts and as construction materials, pulp materials, and food ingredients [7]. Nevertheless, the utilization of bamboo biomass as an industrial material has been limited in certain countries owing to the abundance of other woody biomasses; furthermore, only the bamboo shoot is used as a food ingredient [8]. In addition, a massive amount of lignin is produced during bamboo biomass utilization, which is not as easily usable as the lignin derived from other woody biomasses [9]. Several methods are used to isolate lignin from different plant sources [10]. These processes can be divided into two categories. Lignin is isolated either by removing non-lignin or lignin components. In the lignin extraction process, lignin is produced by dissolving and breaking down biomasses into pieces and isolating and removing the solid residues. Moreover, lignin can also be obtained as a solid residue after selective polysaccharide hydrolysis, along with certain carbohydrate breakdown products. Lignin extraction processes include all pulping processes, such as Kraft, sulfite, soda, and organosolv, whereas selective polysaccharide hydrolysis processes include the dilute acid hydrolysis of lignocellulose to yield sugar monomers, furfural, and levulinic acid [11,12]. However, organosolv fractionation can maximize lignin and minimize its condensation [13]. Previous studies on lignin whitening have mainly focused on TYR inhibition, a whitening-related enzyme. Several studies have reported the use of lignin and its fractions for cosmetic applications. For instance, the lignin obtained from corn stalk using organic solvent extraction was separated into four fractions, and their anti-TYR activities were compared [14]. Another study reported the anti-TYR activity of six lignin samples isolated from three lignocellulosic biomasses via alkali and ethanol organosolv processes [15]. However, lignin and lignin fractions have not been widely used as cosmetic ingredients owing to lignin heterogeneity and the lack of in vitro and in vivo safety and efficacy data.

Therefore, in this study, domestic bamboo (*Phyllostachys nigra* var. henosis) was pretreated using steam explosion, along with stepwise organosolv fractionation. All of the fractionated lignins were characterized using gel permeation chromatography (GPC), Fourier-transform infrared spectroscopy (FT-IR), pyrolysis gas chromatography/mass spectrometry (Py-GC/MS), and 2D heteronuclear single quantum coherence (HSQC). To the best of our knowledge, this is the first study to evaluate the TYR inhibitory activity of lignin fractions at the cellular level for the future development of natural whitening agents in cosmetics. In addition, we report the antioxidant and TYR inhibition kinetics of lignin extracted via steam explosion and subsequent organic solvent fractionation, as well as the structural analyses of the lignin fractions.

## 2. Results

### 2.1. Fractionation and Characterization of Steam-Exploded Bamboo Lignin

#### 2.1.1. Separation Yield and Molecular Weight

In this study, steam-exploded lignin (SEL) was classified into four fractions via sequential extraction with three organic solvents (Figure 1). The yields of the lignin fraction samples were determined and tabulated (Table 1). Ethyl acetate was used as the first solvent to dissolve SEL, and 0.27% total lignin was dissolved (ethyl acetate-soluble SEL fraction; SEL-F1). Subsequently, the ethyl acetate-insoluble residue was sequentially dissolved in methanol to obtain the second fraction (methanol-soluble SEL fraction; SEL-F2) with a 36.43% yield. Acetone was used as the last organic solvent to extract the third fraction (acetone-soluble SEL fraction; SEL-F3, 1.97% SEL). After sequential extraction with the three organic solvents, the remaining 60.33% SEL (insoluble in all three organic solvents) was classified as the last fraction (acetone-insoluble SEL fraction; SEL-F4). GPC was conducted to analyze the molecular weight distributions of the four lignin fractions, and the obtained chromatography curves were normalized (Table 1). The average molecular weight of the original SEL was 4118 g/mol. After fractionation, SEL-F3 had the lowest molecular weight (3230 g/mol) among the four fractions, whereas the insoluble fraction SEL-F4 exhibited the highest molecular weight (3980 g/mol). A previous study reported that the molecular weight of lignin extracted from *Phyllostachys pubescens* using the mill wood treatment was 6424–7754 Da. The extracted lignin structure and molecular weight varied depending on the bamboo type and extraction method used [16]. The lignin polymers featured monomer units arranged in chains of different lengths because all of the polydispersity index (PDI) values exceeded 1. Next, the distribution of each unit (such as guaiacyl [G], syringyl [S], and p-hydroxyphenyl [H]) in the lignin chain was determined to identify the lignin structure.

#### 2.1.2. FT-IR

FT-IR is a useful approach for studying various functional groups as well as the conformational properties of lignin. The FT-IR spectra of SEL, SEL-F1, SEL-F2, SEL-F3, and SEL-F4 are shown in Figure 2. A wide absorption band at 3360 cm^−1^ was assigned to the O–H stretching vibration in the aromatic and aliphatic –OH groups, whereas the bands at 2918 and 2849 cm^−1^ originated from the symmetric and asymmetric C–H vibrations in the methyl and methylene groups. The band at 1699 cm^−1^ indicated the carbonyl stretching in both the conjugated substituted aryl and non-conjugated ketones. The peaks at 1600, 1516, and 1425 cm^−1^ were assigned to the aromatic skeleton vibrations, indicating the primary lignin structure. In addition, the relative band intensities for the aromatic skeleton vibrations assigned at 1600, 1510, and 1425 cm^−1^ were similar, indicating a similar core structure of the lignin fractions. The strong band at 1455 cm^−1^ was attributed to the asymmetric C–H vibrations (asymmetric in the methyl, methylene, and methoxyl groups). The bands at 1326, 1219, and 1113 cm^−1^ were attributed to the S structures in the lignin molecules, whereas the bands at 1265 and 1029 cm^−1^ were associated with the G units in the lignin molecules. The band at 1326 cm^−1^, corresponding to the S and condensed G absorptions, and the G ring breathing with C–O stretching at 1265 cm^−1^ were present. A remarkable characteristic of lignin is the presence of a strong band at 1113 cm^−1^ for the S structures. Aromatic C–H in-plane deformation of the G type was observed at 1029 cm^−1^, whereas the signal of the aromatic C–H out-of-plane deformation appeared at 823 cm^−1^. Careful examination of lignin fractions from the solvent fraction process revealed a significant difference in the absorption at 1699 cm^−1^, corresponding to conjugated ester linkage stretching, which was attributed to the ester bond of the phenolic acids to lignin [17]. This result indicated that the low-molecular-weight lignin fraction contained excess phenolic acids than the high-molecular-weight lignin fraction [18].

#### 2.1.3. Py-GC/MS

The pyrograms of the SEL and the four lignin fractions are shown in Figure 3. Table 2 lists the main group types and retention periods of the 34 compounds detected by Py-GC/MS. Structurally, lignin is a three-dimensional amorphous biopolymer biosynthesized by the radical polymerization of monolignols (coniferyl, sinapyl, and *p*-coumaryl alcohols) to produce G, S, and H units, respectively [19]. In this study, several types of G, S, and H phenol products were detected. The SEL and lignin fractions showed similar peaks and diagnostic compounds, respectively (Figure 3 and Table 2). The main ones were those of 2-methoxy-phenol (peak 4), 2,3-dihydro-benzofuran (peak 6), 2-methoxy-4-vinylphenol (peak 9), 2,6-dimethoxy-phenol (peak 11), propanoic acid (peak 12), 4-hydroxy-2-mercaptopteridine (peak 19), 4-ethenyl-2,6-dimethoxy-phenol (peak 20), 1-(3,5-dimethoxy-4-oxidanyl-phenyl)-ethenone (peak 26), and bis-(2-ethylhexyl)-phthalate (peak 30). The summed peak areas were normalized to 100%, and the total pyrolysis products were classified into four types (G, S, H, and others). Figure 4 shows the percentages of the four types of SEL and lignin fractions. Previous studies have reported that low-molecular-weight polymers have a high content of S units and a low content of G units [17,19]. Wang et al. reported that pine (softwood), poplar (hardwood), and corn stalk (Gramineae plant) were used to extract lignin through organosolv and alkali treatment, and the G:S:H ratio of the extracted lignin was measured using Py-GC/MS [15]. The three types of lignin were pine, poplar, and cornstalk. The GSH composition of pine, poplar, and corn stalk lignin was G_83.05_S_0_H_16.95_, G_48.12_S_38.67_H_13.21_, and G_40.16_S_22.00_H_37.94_, respectively. This study reports that the GSH ratio of SEL, SEL-F1, SEL-F2, SEL-F3, and SEL-F4 were G_18.75_S_35.70_H_17.16_, G_30.73_S_22.33_H_3.38_, G_28.02_S_30.92_H_13.67_, G_11.93_S_29.35_H_20.35_, and G_25.83_S_32.96_H_13.54_, respectively (Figure 3). Thus, the GSH ratio of lignin varies with its source and extraction method. In this study, extraction was carried out after steam-explosion treatment instead of lignin organosolv and alkali treatment. The structures of the G, S, and H units were decomposed, and various structures other than G, S, and H units were confirmed (Figure 4).

#### 2.1.4. NMR Analysis

The chemical structures of SEL and the four fractions (SEL-F1, SEL-F2, SEL-F3, and SEL-F4) were analyzed by ^1^H-, ^13^C-, and 2D HSQC. Lignin properties and structures were confirmed by NMR analysis, and the results are shown in Table 3. The 2D HSQC results confirmed that SEL and the four fractions showed many structural differences. In particular, only p-coumarate and phenylcoumaran structures were confirmed in SEL-F4, and only the ketone and carbon chain structures were observed.

The SEL-F1, SEL-F2, and SEL-F3 samples exhibited a relatively high proportion of methoxyl in the side-chain regions (δC/δH, 56.2/3.04–3.76). In addition, two phenolic acid structures, *p*-coumaric acid and ferulic acid, were identified in the HSQC spectra [19]. SEL-F3 is similar to SEL-F2, but with a different structure. Moreover, SEL-F2 had more carboxyl group C=C bonds and C–O bonds than other lignins. In addition, high methoxy bonding was confirmed, which is consistent with the high antioxidant activity of SEL-F2. Structural analysis indicated that the monomers constituting the lignin dissolved in each organic solvent were different. Due to the differences in the monomer compositions and structures, the antioxidative and TYR inhibition activities of each fraction were different. In the aromatic region, the main cross-signals from the S, G, and H units were observed in all spectra of the lignin samples. The lignin used in this study was GSH-type lignin, in agreement with the fact that the lignin sample originated from bamboo stems [20]. The structures of mill wood lignin (MWL) extracted from the *P. pubescens* species include *p*-coumaric acid, β-*O*-4 aryl ethers, O-4 substructures, β-D-xylopyranoside, 2-O-acetyl-β-D-xylopyranoside, 3-O-acetyl-β-D-xylopyranoside, phenylcoumarans, spirodienones, α,β-diaryl ethers, and p-hydroxycinnamyl. They form structures such as p-coumarates, ferulates, H, G, S units, and benzyl ether [2]. The alkali-extracted lignin from Neosinocalamus affinis consisted of β-*O*-4 linkages, γ-acetylated β-*O*-4 substructures, γ-*p*-coumaroylated β-*O*-4 phenylcoumarane, spirodienone, *p*-hydroxycinnamyl alcohol, H, G, S, and oxidized S units [21]. In this study, the structures of G, S, and H units were confirmed using the same method used in previous studies, and the structures of benzofuran, vinylphenol, and dimethoxy-phenol were confirmed.

### 2.2. Antioxidant Activity of Lignin Fractions

ROS produced by UV radiation, a major factor in melanogenesis, promotes melanin pigment formation in skin cells. Scavenging radicals significantly inhibit melanin formation [21]. In this study, the antioxidant activities of SEL and the four fractions were compared for 2,2-diphenyl-1-picrylhydrazyl (DPPH), 2,2-azino-bis(3-ethylbenzthiazoline-6-sulfonic acid) (ABTS), hydroxyl, and Fe^2+^ chelate radical scavenging activities (Table 4). The extract concentrations were 100–1000 μg/mL and the half-maximal inhibitory concentration (IC_50_) for each antioxidant activity was measured (Figure 5). Among the four lignin fractions, SEL-F2 showed the highest antioxidant activity, except for DPPH scavenging activity. These phenolic structures are presumed to have high antioxidant activity, and the structure and activity of lignin were confirmed to be different depending on the extraction and separation methods. Since various phenolic structures exist in lignin, it shows high antioxidant activity, and the activity varies depending on the extraction method and diverse structures of the lignin. The IC_50_ values of SEL-F2 for DPPH, ABTS, hydroxyl, and Fe^2+^ chelate-scavenging activities were 466.02 ± 4.69, 120.60 ± 2.16, 232.46 ± 13.05, and 73.41 ± 3.22 μg/mL, respectively (Figure 5). The total phenol content (TPC) of SEL-F2 was 578.58 ± 4.30 μg/mL, which was the highest. Consequently, this increased the antioxidant activity of SEL-F2. A previous study reported that the antioxidant activity is closely related to the number of phenolic compounds and directly related to radical scavenging activity [22]. Component analysis revealed that coumaric acid, vanillin, and vanillic acid produced after the decomposition of rice husk lignin do not react with DPPH [23]. However, in this study, after the steam explosion, the lignin was extracted using various organic solvents (ethyl acetate, methanol, and acetone), and several structures, such as 2,6-dimethoxy-phenol, were detected.

### 2.3. Tyrosinase Inhibition Kinetics of SEL-F2

The results in Figure 4 illustrate that the TYR activity was inhibited after incubation with SEL-F2 samples with anti-TYR activity. The non-homogeneity of lignin, including molecular weight distributions and functional group contents, is an important factor affecting its bioactivity, as described in previous reports.

A possible mechanism of TYR inhibition by lignin can be proposed due to their similar chemical structures. Previous studies have reported that lignin competitively reacts with TYR to form a lignin/oxytyrosinase complex, which inhibits the reaction of TYR with 3,4-dihydroxy-L-phenylananine (L-DOPA) and then decreases TYR-mediated L-DOPA oxidation. In this process, the interaction between phenolic OH and copper is critical for lignin/oxytyrosinase complex formation. Thus, the phenolic OH content of lignin is an important factor affecting anti-TYR activity, which agrees with the conclusion that the phenolic OH content of lignin increases with decreasing molecular weight [15]. This confirms that the phenolic OH content was favorable for improving the anti-TYR activity of lignin. The results of the enzyme kinetics experiment showed that SEL-F2 inhibited TYR activity by a complex mechanism, including both competitive and non-competitive inhibitory mechanisms (Figure 6). A previous study conducted the kinetic analysis of lignin and reported that non-competitive inhibition was involved, similar to the results of this study [14]. Mixed non-competitive inhibitors can attach substrates and inhibitors to different binding sites of the enzyme regardless of who attaches first. Lignin acts as a mixed non-competitive inhibitor because covalent and non-covalent bonds coexist and lignin monomers are diverse. To exert in vivo TYR inhibition activity, an appropriate tool to facilitate transdermal penetration of the lignin fraction should be devised. To verify the in vivo safety and efficiency of SEL-F2, a cell line study should be conducted.

### 2.4. Inhibition of Melanin Synthesis in B16F10 Cell Line

Since acid/base and organic solvents are used in lignin fractionation, the toxicity (and/or safety) evaluation of the lignin fraction is very important. The cytotoxic effects of the lignin fractions were investigated using dimethyl sulfoxide (DMSO)-solubilized lignin fractions and the mouse cell line B16F10 to predict their safety levels. Up to 100 μg/mL SEL-F2 treatment for 24 h had no cytotoxicity (Figure 7). Gramineae organosolv lignin, isolated from corn stalk using the ethanol organosolv process, presented the highest anti-TYR activity [23]. In other studies, four fractions of lignin extracted with ethanol showed different anti-TYR activities. Among them, the dichloromethane fraction had the highest TYR inhibition rate (75.12% in 0.5 mg/mL) [13]. As described above, the anti-melanogenic activity of lignin was mainly tested for TYR inhibitory activity; however, unlike previous studies, this study verified TYR inhibition using B16F10 cells. The melanin content experiment showed that whitening activity decreased dose-dependently 48 h after treatment with 50–100 μg/mL SEL-F2 and 1 μg/mL α-MSH [24].

### 2.5. Effect of SEL-F2 on Anti-Melanogenesis-Related Proteins in B16F10 Cells

Melanogenesis mainly depends on regulating melanogenic proteins such as TYR, TRP-1, and TRP-2. Western blotting was performed to determine whether SEL-F2-mediated inhibition was related to melanogenesis-related protein regulation. TYR, TRP-1, and TRP-2 protein expression levels were increased by α-MSH treatment, whereas SEL-F2 significantly decreased TYR, TRP-1, and TRP-2 protein expression levels in B16F10 cells (Figure 8a,b).

The expression of tyrosinase-related proteins, TYR, TRP-1, and TRP-2, decreased in a concentration-dependent manner. In a previous study, Wu et al. separated three new lignin glycosides from *Castanea ehnryi* and two known lignins [25]. The structure of the isolated lignin was mainly 2,3-dihydro-2-[4-(β-glucopyranosyl (1,2)-[β-glucopyranosyl(1,6)]-β-glucopyranosyloxy)-3-methoxyphenyl]-3-(hydroxymethyl)-7-methoxy5-benzofuranpropanol and it was reported that all of them inhibited tyrosinase. In another study, six lignin samples (obtained by alkali and ethanol extraction from three typical lignocellulosic materials) showed excellent tyrosinase inhibitory effects [15].

Microphthalmia-associated transcription factor (MITF) is expressed in various tissues and is a basic helix-loop-helix leucine zipper transcription factor [25,26]. MITF is known to mediate the differentiation effect of α-MSH by regulating enzymes that are essential for melanogenesis in differentiated melanocytes [27]. MITF is a transcription factor that performs an important function during melanin synthesis. In this experiment, SEL-F2 treated with B16F10 melanoma cells showed a reduced MITF expression in a concentration-dependent manner (Figure 9).

PKA is a cyclic AMP (cAMP)-dependent protein kinase that regulates the activity of various enzymes and is involved in the cell cycle and proliferation. PKA increases tyrosinase activity and induces phosphorylation as well as the activity of tyrosinase. Therefore, when PKA expression is suppressed, melanin production is suppressed by preventing tyrosinase activation. As a result of the experiment, SEL-F2 showed inhibitory effects at concentrations of 25, 50, and 100 μg/mL (Figure 10).

## 3. Discussion

Based on our data, the signaling pathways of in vivo TYR inhibition by SEL-F2 were proposed (Figure 11). When exposed to UV rays, the skin defends itself from being affected by them, and skin darkening is a UV defense mechanism. UV radiation triggers melanogenesis through the melanocortin-1 receptor (MC1R), which is activated by α-MSH [28]. TYR, a key enzyme in melanin synthesis, oxidizes tyrosine and L-DOPA. In addition, TYR-related protein-1 (TRP-1) and TYR-related protein-2 (TRP-2), which reside in melanosomes, regulate melanin production [28]. The expression of these enzymes is regulated by microphthalmia-associated transcription factor (MITF), whose expression and activity are regulated by the protein kinase A (PKA) and cAMP-response element-binding protein (CREB). Therefore, when PKA and CREB expression is suppressed, melanin production is suppressed by preventing TYR activation. In this study, we demonstrated the anti-melanogenesis mechanism of SEL-F2 via the PKA/CREB protein signaling pathway. UV rays stimulate pigment-forming cells (called melanocytes) in the basal cell layer deep in the skin to produce brown melanin particles. Therefore, using sunscreen can prevent skin irritation from UV rays and keep the skin clear and transparent. Typical sunscreens include zinc oxide, titanium oxide, and oxybenzone. However, they have side effects such as cloudiness, skin irritation, and cell DNA damage [10]. Natural sunscreens have recently been in the spotlight because of these side effects; moreover, as lignin has been confirmed to have a strong antioxidant function, studies on the effects of sunscreen are being actively conducted [29]. Since lignin has many phenolic and ketone groups, the excitation state of π electrons that absorb ultraviolet light may be stabilized, and through these properties, a wide range of light can be absorbed [30]. In addition, the phenolic group of lignin is useful while mixing it with other aromatic compounds because it can further stabilize the electrons in the excited state by π-π* stacking with other aromatic rings [31]. In a previous study, the molecular weight of Kraft-treated lignin was 3000–7000 Da, while in another study, organosolv-treated lignin had a molecular weight of 3000–5000 Da, demonstrating that both lignins have a UV-protection effect [13]. Wang et al. reported that the molecular weight of lignin extracted from the wheat stalk (WSL) is 1090–1150 Da. WSL satisfactorily ameliorated inflammation and oxidative stress in RAW 264.7 cells and colitis mice by activating NrSEL-F2 and suppressing NFκB signaling [32]. Lignin is mainly extracted using an organic solvent and alkali. Additionally, hydrophilic and lipophilic substances are extracted. The molecular weight of lignin varies depending on the extraction solvent or extraction method (physical, chemical, and biological). Macromolecules with a molecular weight of 1000 Da or more are hydrophilic and, thus, exhibit various physiological activities when applied to cells.

In general, approximately 500 Da molecules penetrate the skin, which is the required molecular size for transdermal absorption of the skin [33]. This study confirmed that the molecular weight of lignin extracted with alkali after steam explosion treatment is 3200–4000 Da or less. Using such high-molecular-weight lignin as a sunscreen or whitening agent may promote absorption through the skin by preparing a nano-emulsion formulation of 500 Da or less. Therefore, lignin treated with alkali after steam bombardment is a suitable material for whitening cosmetics.

## 4. Materials and Methods

### 4.1. Chemicals

DPPH, sodium hydroxide (NaOH), ABTS, FeCl2, ferrozine (for ferrous metal ion chelation), Folin-Ciocalteu reagent, sodium carbonate (for total phenolics), tetrahydrofuran (THF), DMSO-*d*_6_, mushroom TYR, 3-(4,5-dimethylthiazol-2-yl)-2,5-diphenyltetrazolium bromide (MTT), and Tween-20 were obtained from Sigma-Aldrich (Saint Louis, MO, USA). Antibodies against TYR, TRP-1, TRP-2, MITF, PKA, and CREB were obtained from Santa Cruz Biotechnology (Dallas, TX, USA). The phospho-PKA and phospho-CREB antibodies were obtained from Cell Signaling Technology (Danvers, MA, USA). α-MSH, DMSO, sodium nitrite, and an antibody against β-actin were obtained from Sigma-Aldrich. Horseradish peroxidase-conjugated goat anti-mouse and anti-rabbit antibodies were purchased from Invitrogen (Carlsbad, CA, USA). Phosphate-buffered saline (PBS) and Dulbecco’s modified Eagle’s medium (DMEM) supplemented with 10% fetal bovine serum (FBS) and penicillin were obtained from HyClone (Cytiva, MA, USA). All other chemicals and reagents used were of molecular biology grade and commercially available.

### 4.2. Bamboo Biomass and Lignin Preparation

The *P. nigra* var. henosis used in this study was obtained from a local supplier in Damyang Province, Korea (GPS position 35.2909868° N, 126.9844976° E). The fresh stems were dried overnight at 60 °C, sliced into small pieces (1–2 cm), and ground into powder. Then, 1 kg bamboo stem powder was soaked in distilled water for 24 h at 25 °C, followed by a steam explosion treatment for processing the lignocellulosic materials using a steam explosion machine (SPE-STE-40K, ILSHIN AUTOCLAVE CO., LTD; Daejeon, Korea) at 210 °C and 2.0 MPa for 5 min [34]. The exploded material was collected and oven-dried at 40 °C for 24 h. The scheme for alkali extraction of lignin fractions from steam-exploded samples is illustrated in Figure 1. The steam-exploded samples were extracted with 0.6% aqueous NaOH solution at 80 °C for 3 h with a 1:20 solid to liquor ratio (g/mL). The alkali-extracted filtrates were neutralized to pH 5.5 with H_2_SO_4_ and evaporated at reduced pressure. Then, 600 mL ethyl acetate, methanol, and acetone were used for serial extractions at 500 rpm and room temperature for 3 h. Therefore, the total SEL was further separated into five fractions: SEL, SEL-F1, SEL-F2, SEL-F3, and SEL-F4 (Figure 1).

### 4.3. GPC Analysis

GPC measurement was performed as reported previously [35]. The number-average molecular weight (*M*_n_), weight-average molecular weight (*M*_w_), and PDI (*M*_w_/*M*_n_) of the lignin fractions were determined using the GPC apparatus (EcoSEC HLC-8320 GPC, Tosoh, Japan). The lignin molecular-weight distributions were determined on an instrument (1200 series, Agilent Technologies; Santa Clara, CA, USA) equipped with an RI detector and a 2 × TSK gel Supermultipore HZ-M + TSK gel SuperHZ-2500 gel permeation column (4.6 × 150 mm^2^). Polystyrene was used as the standard to determine the molecular weight of the lignin. The lignin samples (3 mg) were diluted in 1 mL THF, and 30 μL samples were injected. The column was operated at 40 °C and eluted using THF at a 0.35 mL/min flow rate.

### 4.4. Physico-Chemical Characterization of Lignin

The purity of the lignin samples was analyzed using a modified component determination method from the National Renewable Energy Laboratory [36]. The FT-IR spectra of the total SEL and its fractions were obtained using an FT-IR spectrophotometer (IRAffinity-1S, Shimadzu, Japan) using the MIRacle technique. Specific IR absorption was measured within 4000–400 cm^−1^ with a 2 cm^−1^ resolution. Py-GC/MS was conducted using a pyrolyzer (Agilent Technologies 7890 B and 5977 B MDS system) with directly connected gas chromatography and mass spectrometry. Py-GC/MS analysis was conducted using a Frontier Lab Single-Shot Pyrolyzer (Py-2020iS, Frontier Lab; Fukushima, Japan) in addition to Agilent Technologies 7890 B, 5977 B MDS system concatenated to a gas chromatograph, as well as being equipped with a mass spectrometer, according to the reported procedure. Approximately 100 μg lignin sample was pyrolyzed in a quartz tube at 600 °C for 12 s under helium as the carrier gas with 1 mL/min mean linear velocity. 2D heteronuclear single quantum coherence (HSQC) correlation NMR spectra were obtained on a Bruker AVIII 400 MHz spectrometer (Bruker Corporation; Billerica, MA, USA) operated at a 400 MHz frequency. The samples (70 mg each) were dissolved in 0.5 mL DMSO-*d*_6_.

### 4.5. Antioxidant Activity Assay

To measure the antioxidant activity, each lignin fraction was DPPH-free. Radical scavenging activity [37], ABTS radical scavenging activity [38], and hydroxyl radical scavenging methods [39] were used. The chelation of ferrous ions by SEL, all SEL fractions, and standard molecules was estimated using a previously described method [40]. Briefly, SEL and SEL-F1–F4 (0.4 mL, 25–50 μg/mL) were added to 0.05 mL 2 mM FeCl_2_. The reaction was initiated by adding 0.2 mL solution containing 5 mM ferrozine, and the total volume was adjusted to 4 mL using ethanol. The mixture was shaken vigorously and incubated at room temperature for 10 min. The absorbance of the solution was measured spectrophotometrically at 562 nm. The inhibition percentage of ferrozine-Fe^2+^ complex formation was calculated using the following formula: metal-chelating effect (%) = [(A_Control_ − A_Sample_)/A_Control_] × 100, where A_Control_ is the absorbance of the control, and A_Sample_ is the absorbance in the presence of either caffeic acid or standards. The control contained FeCl_2_ and ferrozine, which are complex formation molecules [41].

### 4.6. Total Polyphenol Content

The TPC of the fractionated samples was measured using a modified Folin-Ciocalteu method [42]. A 0.5 mL extract aliquot was mixed with 0.5 mL 2% (*w*/*v*) sodium carbonate and 0.5 mL Folin-Ciocalteu reagent. After incubation at room temperature for 30 min, the absorbance of the reaction mixture was measured at 750 nm using a UV-Vis spectrophotometer (S-3100, SCINCO; Seoul, Korea). The extract samples were evaluated at a 1 mg/mL final concentration. The TPC was expressed as p-coumaric acid equivalent (mg/mL) using the following equation based on the calibration curve: y = 2.9533x + 0.1146, R^2^ = 0.9883; where x is the absorbance and y is the *p*-coumaric acid equivalent (μg/g).

### 4.7. Anti-Melanogenic Activity Assay

#### 4.7.1. Tyrosinase Inhibition Assay and Enzyme Kinetics Assay

The TYR inhibition experiment was performed by partially modifying the experimental method described by Choi et al. [43]. TYR is highly known as a key enzyme that determines the overall oxidative kinetics of converting L-tyrosine to L-DOPA and further to DOPA-quinone. An L-DOPA substrate was used in this study, and experiments were performed at 0.5, 1.0, 1.5, and 2.0 mM.

TYR inhibition activity was detected using a spectrophotometer (Shimadzu UV-2450, Shimadzu Corporation; Kyoto, Japan), and an IC_50_ quantitative assay of TYR was performed according to the method described by Fan et al. [44,45,46]. First, 20 μL aliquots of a solution comprising 1750 U/mL mushroom TYR (Sigma-Aldrich) were added to 96-well microplates. Then, 100 μL PBS (pH 6.8) and 60 μL SEL-F2 (100 μg/mL in 25% DMSO) were added. The absorbance of the wells was measured at 510 nm (T_0_) using a microplate reader (Synergy HT, BIO-TEX; Winooski, VT, USA). Subsequently, the microplates were incubated at 27 ± 1 °C for 30 min, and the absorbance was measured again (T_1_). An additional reaction period of 30 min at 30 ± 1 °C was allowed, and the absorbance was measured again (T_2_). The inhibitory percentages at the two time points (T_1_ and T_2_) were obtained based on the following formula: IA% = [(c − S)/c] × 100, where IA% is the inhibitory activity, C is the negative control absorbance, and S is the sample or positive control absorbance (the absorbance at time T_0_ subtracted from the absorbance at time T_1_ or T_2_).

#### 4.7.2. MTT Assay for SEL-F2

The MTT assay was used to evaluate SEL-F2 cytotoxicity; the method adopted by Choi et al. was modified for this purpose [25]. A cytotoxicity proliferation assay was performed using the MTT assay. B16F10 cells (1 × 10^4^ cells/mL) were cultured in 24-well plates. After 24 h, the cells were treated with 25, 50, and 100 μg/mL SEL-F2 for 48 h. At the end of the incubation period, 100 μL MTT solution (1 mg/mL in DMEM) was added to each well. After incubation at 37 °C for 1 h, the medium was gently removed and 400 μL DMSO was added. The absorbance of each well was then measured at 570 nm using a spectrophotometer.

#### 4.7.3. Melanin Contents in B16F10 Cell Line

B16F10 melanoma cells, forming a murine melanoma cell line, were purchased from ATCC (American Type Culture Collection; Rockville, MD, USA). The cells were maintained in DMEM and supplemented with 10% FBS, 50 U/mL penicillin, and 50 μg/mL streptomycin at 27 °C in a humidified atmosphere with 5% CO_2_ at 37 °C. Melanin content was determined according to a modified method described by Hosoi et al., 1985 [47]. B16F10 cells (1 × 10^5^ cells/well) were cultured in 6-well plates. After 24 h of incubation, the cells were stimulated with 1 μg/mL α-MSH, followed by a treatment with 50 and 100 μg/mL SEL-F2 and 100 μg/mL arbutin. The cells were obtained by washing them with PBS and scraping them using a scraper. The obtained cells were centrifuged at 12,000× *g* rpm for 5 min, then the cell pellet was dissolved in NaOH and lysed at 95 °C for 1 h. Melanin was quantified at 405 nm using 96-well plates.

#### 4.7.4. Immunoblot Analysis

Sodium dodecyl sulfate-polyacrylamide gel electrophoresis and immunoblot analyses were performed for protein extraction and intracellular fractionation [48]. The samples were separated by 7.5% gel electrophoresis and electrophoretically transferred to a nitrocellulose membrane, which was incubated with the indicated primary antibodies, followed by incubation with horseradish peroxidase-conjugated secondary antibodies. The immunoreactive proteins were visualized using enhanced chemiluminescence (ECL, GE Healthcare; Chicago, IL, USA). Equal protein loading was verified by immunoblotting with β-actin.

## 5. Conclusions

In this study, SEL-F2 showed the highest antioxidant and TYR inhibition activities due to differences in monomer content and structure. Even though the molecular weights of SEL-F2 are considerably high (3230 and 3980 g/mol), it strongly inhibited the whitening-related proteins TYR, TRP-1, and TRP-2 at a cellular level. SEL-F2 dissolved in DMSO inhibits the expression of MITF, a key transcription factor related to melanogenesis. Thus, SEL-F2 may be a potential material for whitening and anti-aging cosmetics. In-depth research on cosmetic formulations, such as nano-emulsions with SEL-F2, should be conducted in the future.

## Figures and Tables

**Figure 1 ijms-23-07462-f001:**
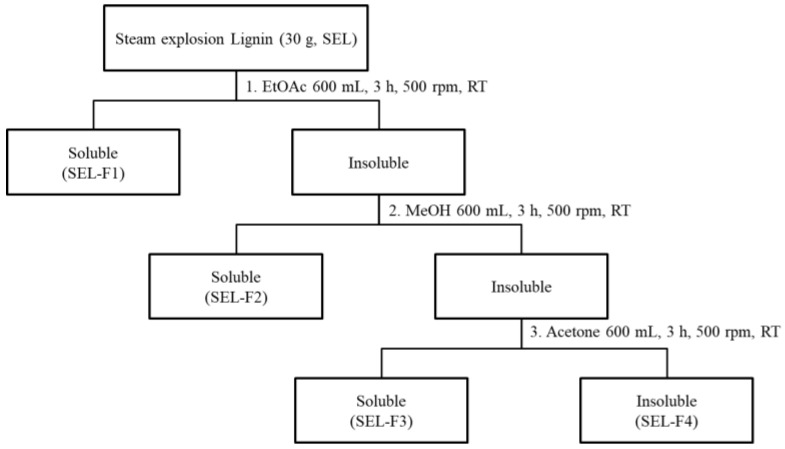
Schematic diagram of the extract preparation from steam-exploded lignin.

**Figure 2 ijms-23-07462-f002:**
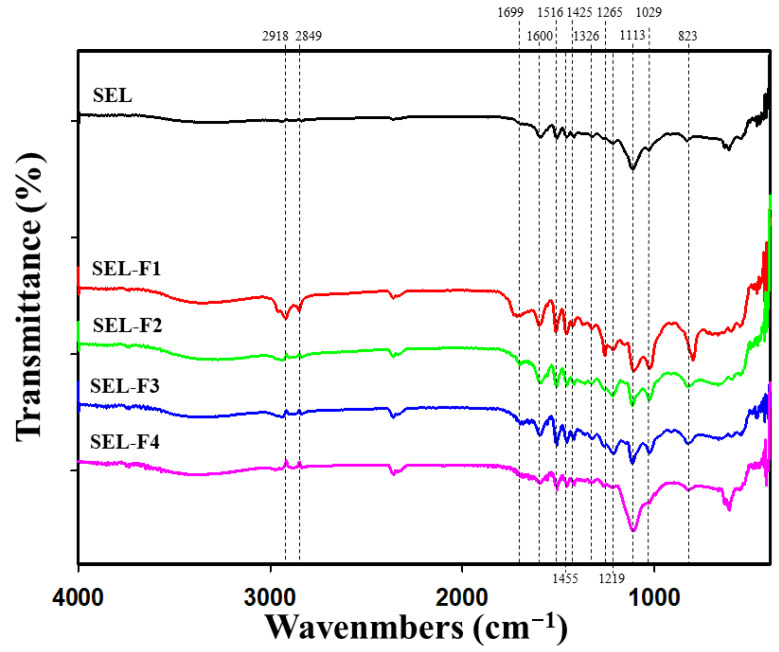
Fourier-transform infrared spectroscopy (FT-IR) spectra of steam-exploded lignin (SEL) and four fractions (SEL-F1, SEL-F2, SEL-F3, and SEL-F4).

**Figure 3 ijms-23-07462-f003:**
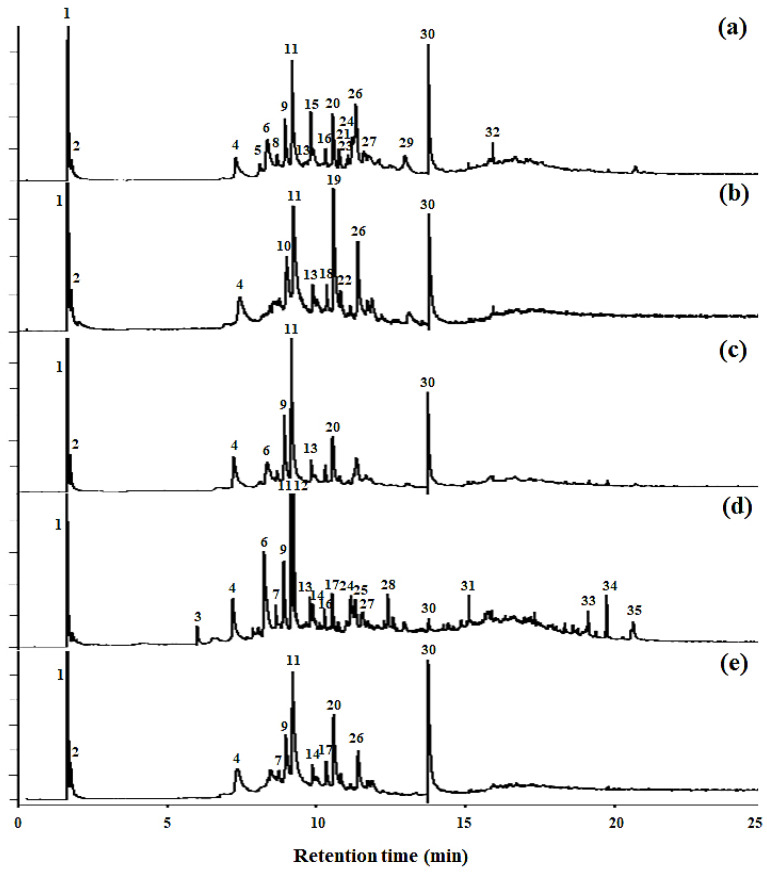
Pyrolysis gas chromatography/mass spectrometry (Py-GC/MS) analysis of steam-exploded lignin (SEL) fractions: (**a**) SEL-F4, (**b**) SEL-F3, (**c**) SEL-F2, (**d**) SEL-F1, (**e**) SEL. For peak identification, refer to Table 2.

**Figure 4 ijms-23-07462-f004:**
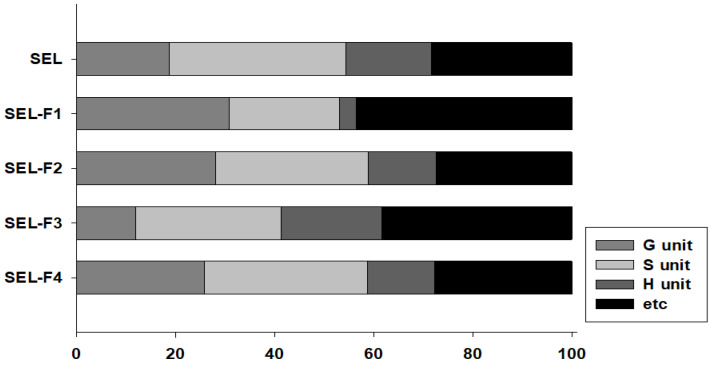
Distribution between guaiacyl (G)-unit, syringyl (S)-unit, p-hydroxyphenyl (H)-unit, and other lignin fraction samples determined by pyrolysis gas chromatography/mass spectrometry (Py-GC/MS).

**Figure 5 ijms-23-07462-f005:**
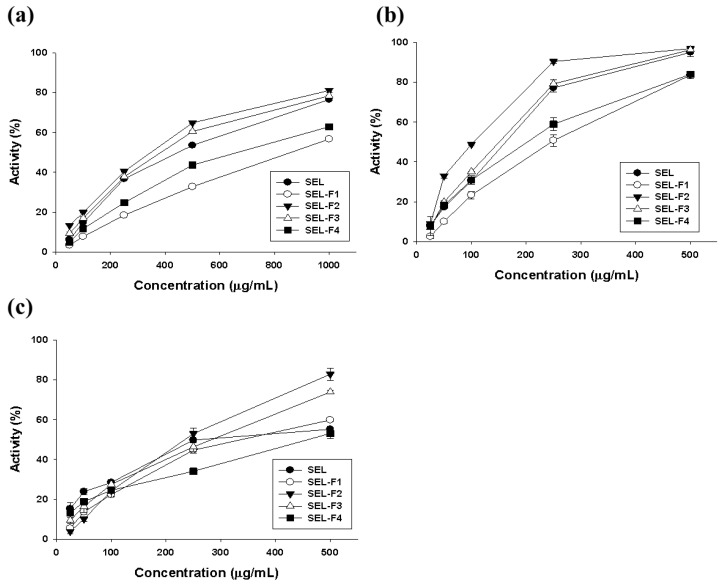
Antioxidant activity results: (**a**) 2,2-diphenyl-1-picrylhydrazyl (DPPH), (**b**) 2,2-azino-bis(3-ethylbenzthiazoline-6-sulfonic acid) (ABTS), and (**c**) hydroxyl radical activities.

**Figure 6 ijms-23-07462-f006:**
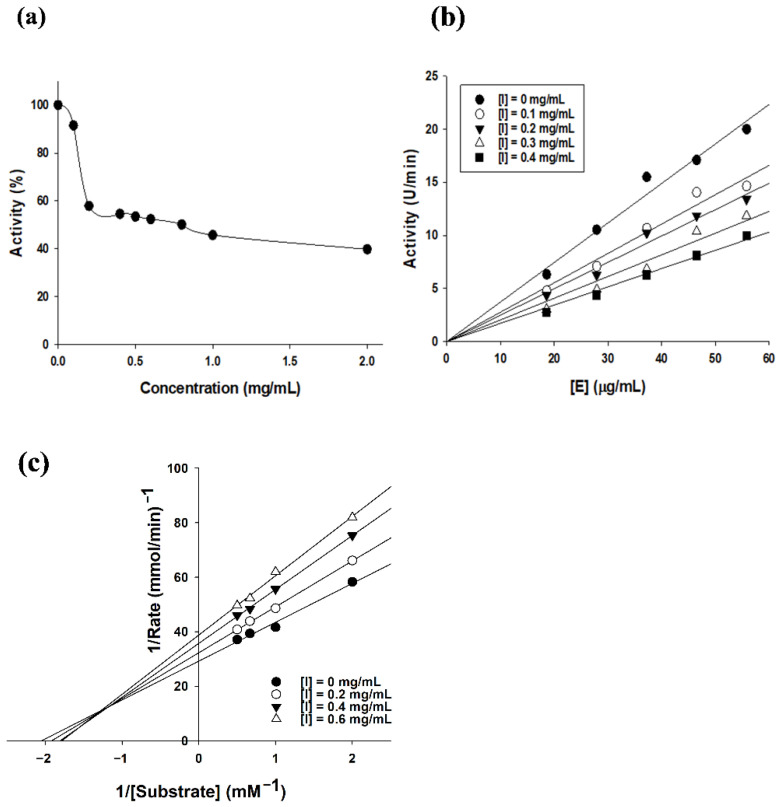
(**a**) Inhibition activity of methanol-soluble steam-exploded lignin fraction (SEL-F2); (**b**) effects of tyrosinase (TYR) concentrations ([E]) on its activity at different SEL-F2 concentrations ([I]); (**c**) TYR inhibition kinetics of SEL-F2 by Lineweaver-Burk plots. The two parts on the right represent the secondary slope and the intercept of the straight lines versus SEL-F2 concentration, respectively, and the fluorescence spectra of TYR in the presence of different SEL-F2 concentrations.

**Figure 7 ijms-23-07462-f007:**
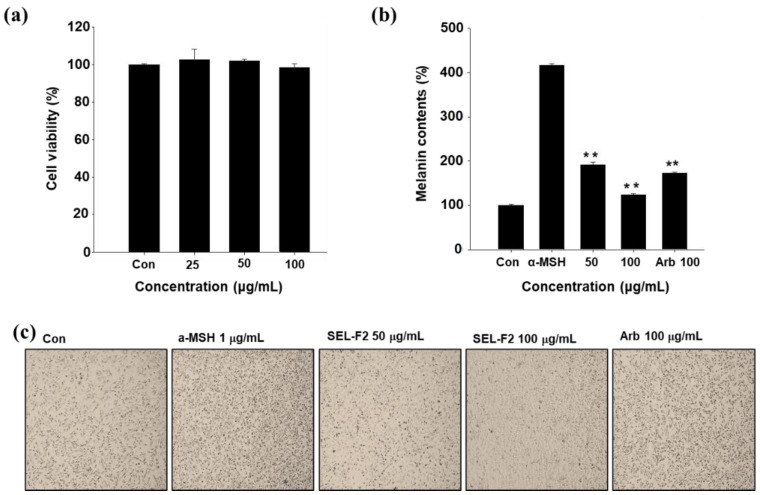
(**a**) Cytotoxicity result (3-(4,5-dimethylthiazol-2-yl)-2,5-diphenyltetrazolium bromide (MTT) assay) of methanol-soluble steam-exploded lignin fraction (SEL-F2). (**b**) Effects of SEL-F2 on melanogenesis and morphological changes in B16F10 cells. (**c**) B16F10 melanoma cells were cultured for 48 h in the presence of 50–100 μg/mL SEL-F2 and 100 μg/mL arbutin as a positive control, or 1 μg/mL α-melanocyte-stimulating hormone (α-MSH). CON: control B16F10 cell line without any treatment. Represents the data of three separate experiments. ** *p* < 0.01; significant compared with α-MSH.

**Figure 8 ijms-23-07462-f008:**
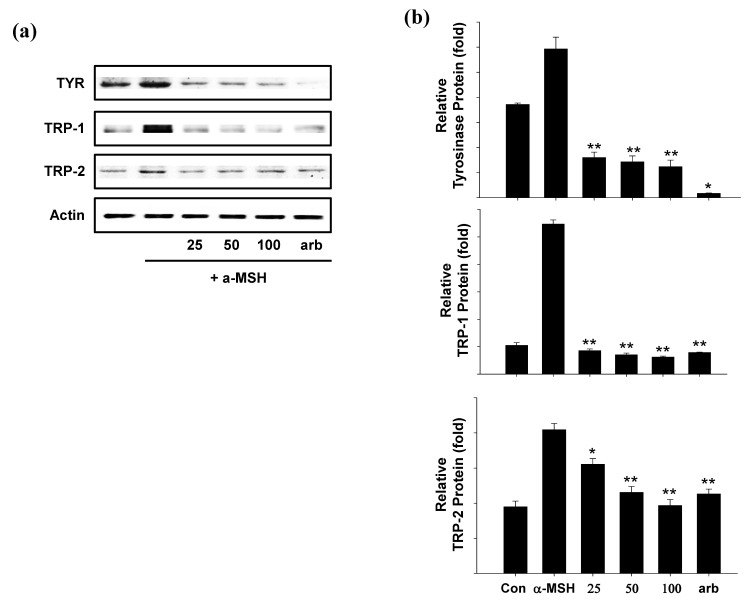
(**a**) Effect of the methanol-soluble steam-exploded lignin fraction (SEL-F2) on tyrosinase (TYR), TYR-related protein 1 (TRP-1), and TYR-related protein 2 (TRP-2) protein expression levels in B16F10 melanoma cells. These cells were treated with 25, 50, or 100 μg/mL SEL-F2 or arbutin prior to α-melanocyte-stimulating hormone (α-MSH) treatment for 24 h. The loading control was assessed using a β-actin antibody; (**b**) quantitative analysis of tyrosinase, (**c**) TRP-1, (**d**) TRP-2 by western blotting. Cell lysates were subjected to western blotting using antibodies against tyrosinase, TRP-1, and TRP-2. * *p* < 0.05, ** *p* < 0.01.

**Figure 9 ijms-23-07462-f009:**
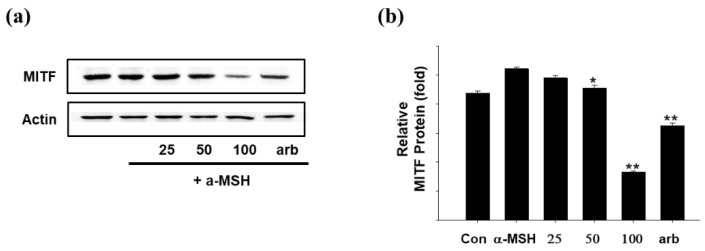
(**a**) Effect of methanol-soluble steam-exploded lignin fraction (SEL-F2) on microphthalmia-associated transcription factor (MITF) protein expression levels in B16F10 melanoma cells. B16F10 melanoma cells were treated with 25, 50, and 100 μg/mL SEL-F2 or arbutin prior to α-melanocyte-stimulating hormone (α-MSH) treatment for 8 h. The loading control was assessed using a β-actin antibody. (**b**) Quantitative analysis by Western blotting. Cell lysates were subjected to Western blotting using antibodies against MITF. * *p* < 0.05, ** *p* < 0.01.

**Figure 10 ijms-23-07462-f010:**
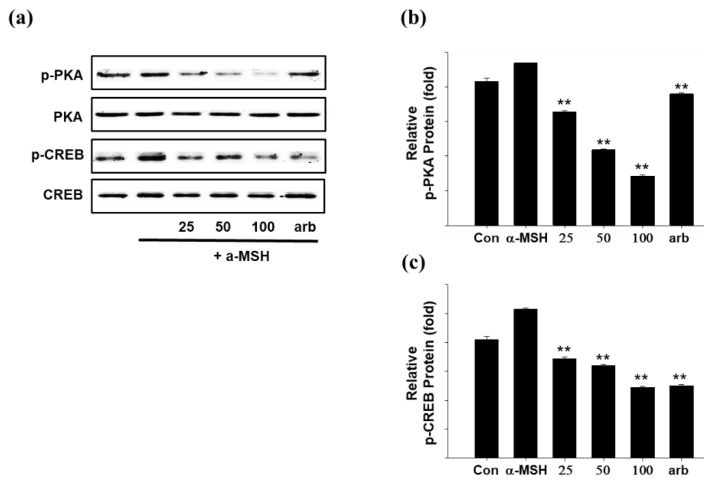
(**a**) Effect of methanol-soluble steam-exploded lignin fraction (SEL-F2) on protein kinase A (PKA), phospho-PKA (p-PKA), cAMP-response element-binding protein (CREB), and phosphor-CREB (p-CREB) protein expression levels in B16F10 melanoma cells. B16F10 melanoma cells were treated with 25, 50, and 100 μg/mL SEL-F2 or arbutin prior to α-MSH treatment for 6 h. The loading control was assessed using a β-actin antibody; quantitative analysis of (**b**) p-PKA and (**c**) p-CREB. Cell lysates were subjected to Western blotting using antibodies against p-PKA, PKA, p-CREB, and CREB. ** *p* < 0.01.

**Figure 11 ijms-23-07462-f011:**
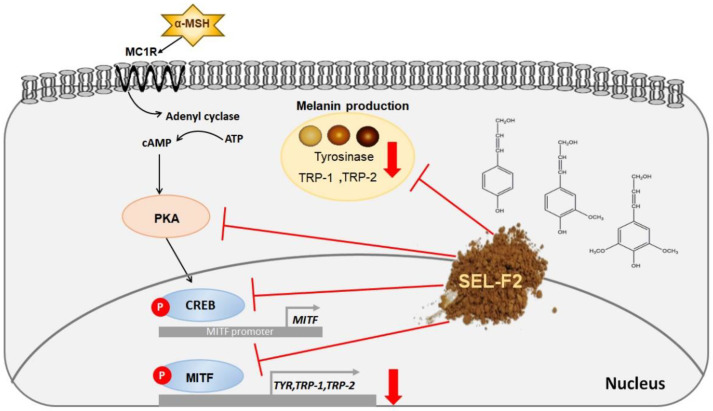
Possible pathways of in vivo tyrosinase inhibition by methanol-soluble steam-exploded lignin fraction (SEL-F2).

**Table 1 ijms-23-07462-t001:** Yield and gel permeation chromatography (GPC) analysis results of steam-exploded lignin (SEL) and SEL fractions 1–4 (SEL-F1, SEL-F2, SEL-F3, and SEL-F4).

Sample	Yield (%)	GPC		
		*M* _w_	*M* _n_	PDI
SEL	-	4118 ± 7.76	2342 ± 24.90	1.759 ± 0.02
SEL-F1	0.27	3512 ± 25.72	2110 ± 8.16	1.664 ± 0.02
SEL-F2	36.43	3914 ± 12.66	2251 ± 17.56	1.739 ± 0.01
SEL-F3	1.97	3230 ± 40.82	2046 ± 19.19	1.579 ± 0.03
SEL-F4	60.33	3980 ± 16.33	2244 ± 13.06	1.774 ± 0.02

*M*_n_: number-average molecular weight; *M*_w_: weight-average molecular weight; PDI: polydispersity index (=*M*_w_/*M*_n_). Each piece of data is the mean ± standard deviation (SD) of three experiments.

**Table 2 ijms-23-07462-t002:** Identity and relative molar abundances of the compounds released after pyrolysis gas chromatography/mass spectrometry (Py-GC/MS) of steam-exploded lignin (SEL) and SEL fractions 1–4 (SEL-F1, SEL-F2, SEL-F3, and SEL-F4).

No.	Compound	SEL	SEL-F1	SEL-F2	SEL-F3	SEL-F4
1	Carbon dioxide	18.85	9.33	15.49	21.11	7.89
2	Acetaldehyde	3.86	-	3.95	3.39	2.38
3	1-Bromo-2-methyl-cyclohexane	-	1.57	-	-	-
4	2-Methoxy-phenol	8.09	8.68	8.26	7.55	3.87
5	4-Methoxy-2-methylphenol	-	-	-	-	1.86
6	2,3-Dihydro-benzofuran	-	14.96	7.95	-	10.24
7	1,5-Heptandien-3-yne	4.5	1.19	-	-	-
8	2-Methyl-benzaldehyde	-	-	-	-	2.27
9	2-Methoxy-4-vinylphenol	8.32	7.06	10.33	-	7.96
10	3-Methoxyacetophenone	-	-	-	8.7	-
11	2,6-Dimethoxy-phenol	21.18	14.56	22.45	18.65	14.68
12	Propanoic acid	-	6.49	-	-	-
13	4,5-Dimethoxy-2-methylphenol	-	1.98	1.75	1.73	5.49
14	3-Methyl-2-propylcyclopent-2en-1-one	1.18	3.49	-	-	-
15	1,5-Heptadien-3-yne	-	-	-	-	3.59
16	1,1’-Ethylidenbis-benzene	-	2.03	-	-	3.21
17	2,4-Dimethoxybenzoic acid	2.34	3.72	-	-	-
18	4-Ethyl-2,6-dimethoxy-phenol	-	-	-	2.16	-
19	4-Hydroxy-2mercaptopteridine	-	-	-	13.87	-
20	4-Ethenyl-2,6-dimethoxy-phenol	8.97	-	8.47	-	6.24
21	2,6-Dimethoxy-4-[(Z)-prop-1-enyl]phenol	-	3.62	7.68	-	3.16
22	2,5-Dimethoxy-4-methyl-benzaldehyde	-	-	-	2.65	-
24	4-Hydroxy-3,5-dimethoxy benzaldehyde	-	5.25	-	-	2.59
25	2,6-Dimethoxy-4-(2-propenyl) phenol	5.55	-	-	8.54	8.43
26	1-(3,5-Dimethoxy-4-oxidanyl-phenyl)ethanone	-	3.04	-	-	1.22
27	Acetamide	-	2.7	-	-	-
28	1,5-Heptadien-3-yne	-	-	-	-	3.53
29	4-4’-(1-Methylethylidene)bis-phenol	17.15	1.34	13.66	11.65	10.33
30	*Bis*(2-Methylhexyl) phthalate	-	1.06	-	-	-
31	4,4’-Methylenebis(2,6-dimethoxy-phenol)	-	-	-	-	1.02
32	β-Progesterone	-	1.89	-	-	-
33	Friedelan-y-al	-	3.51	-	-	-
34	Syringaresinol	-	2.52	-	-	-

**Table 3 ijms-23-07462-t003:** Assignments of heteronuclear single quantum coherence spectroscopy (HSQC) spectra results of steam-exploded lignin (SEL) and SEL fractions 1–4 (SEL-F1, SEL-F2, SEL-F3, and SEL-F4).

Assignments	δ_C_/δ_H_ (ppm)				
	SEL	SEL-F1	SEL-F2	SEL-F3	SEL-F4
C_β_-H_β_ in phenylcoumaran substructures	-	51.6/3.56	-	-	-
C_β_-H_β_ in β-β’ resinol substructures	54.2/3.04	53.2/3.76	54.23/3.04	54.2/3.05	-
C-H in methoxyls	-	56.2/3.74	56.5/3.75	-	56.0/3.97
C_β_-H_β_ in β-*O*-4β’ substructures	-	59.5/3.56	-	-	-
C_β_-H_β_ in g-acylated β-*O*-4β’ substructures	-	60.1/4.01	-	-	-
C_β_-H_β_ in p-hydroxycinnamyl alcohol end groups	-	66.1/3.81	-	63.37/3.38	-
C_β_-H_β_ in β-β’ resinol substructures	71.6/4.17	-	71.6/4.15	71.6/4.17	-
C_β_-H_β_ in phenylcoumaran substructures	-	70.9/4.78	72.6/4.82	72.5/4.86	72.3/4.86
C_β_-H_β_ in β-β’ resinol substructures	85.7/4.61	-	85.8/4.61	85.7/4.62	-
C_β_ H_β_ in β-*O*-4β’ linked to a S unit	86.4/4.11	-	86.5/4.11	86.5/4.12	86.3/4.12
C_β_-H_β_ in phenylcoumaran substructures	-	-	87.6/5.43	87.6/5.44	-
C_2,6_-H_2,6_ in syringyl units	104.4/6.66	-	104.0/6.60	104.3/6.69	-
C_2,6_-H_2,6_ in oxidized (C=O) S units	107.0/7.30	107.0/7.21	107.1/7.29	107.4/7.21	107.1/7.18
C_2_-H_2_ in guaiacyl units	111.7/6.98	-	-	110.9/6.90	110.08/6.89
C_5_-H_5_ in guaiacyl units	116.1/6.77	-	116.1/6.76	116.1/6.79	116.12/6.76
C_6_-H_6_ in guaiacyl units	119.1/6.74	-	119.1/6.75	119.1/6.75	-
C_2,6_-H_2,6_ in H units	127.7/7.15	-	-	-	-
C_3,5_-H_3,5_ in *p*-coumaric acid	115.4/6.51	116.2/6.79	-	116.4/6.31	-
C_2,6_-H_2,6_ in *p*-coumaric acid	130.0/7.41	130.6/7.50	129.6/7.37	-	130.7/7.41
C_7_-H_7_ in *p*-coumaric acid	141.3/7.32	144.7/7.47	140.2/7.27	144.2/7.49	-
C_8_-H_8_ in *p*-coumaric acid	120.3/6.27	-	-	116.5/6.28	-

G, guaiacyl; H, *p*-hydroxyphenyl; S, syringyl.

**Table 4 ijms-23-07462-t004:** Antioxidant activity results and total polyphenol contents of steam-exploded lignin (SEL) and SEL fractions 1–4 (SEL-F1, SEL-F2, SEL-F3, and SEL-F4).

Sample	IC_50_ (μg/mL)			Fe^2+^ Chelate (%)	TPC (μg/mL)
	DPPH	ABTS	Hydroxyl Radical		
SEL	439.06 ± 1.90	161.33 ± 4.33	249.92 ± 5.21	61.57 ± 2.53	512.23 ± 4.22
SEL-F1	757.24 ± 10.81	241.89 ± 10.46	372.50 ± 1.27	49.51 ± 1.48	395.74 ± 8.82
SEL-F2	466.02 ± 4.69	120.60 ± 2.16	232.46 ± 13.05	73.41 ± 3.22	578.58 ± 4.30
SEL-F3	392.09 ± 3.08	154.40 ± 2.69	266.81 ± 15.88	65.02 ± 4.21	516.28 ± 6.87
SEL-F4	567.12 ± 0.86	205.51 ± 12.86	429.95 ± 3.72	53.16 ± 3.36	448.33 ± 4.14

ABTS: 2,2-azino-bis(3-ethylbenzthiazoline-6-sulfonic acid); DPPH: 2,2-diphenyl-1-picrylhydrazyl; IC_50_: half maximal inhibitory concentration; TPC: total phenol content. Each piece of data is the mean ± standard deviation (SD) of three experiments.

## Data Availability

Not applicable.

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
