# Peer review of "Bamboo Lignin Fractions with In Vitro Tyrosinase Inhibition Activity Downregulate Melanogenesis in B16F10 Cells via PKA/CREB Signaling Pathway"

_ijms, 2022, doi:10.3390/ijms23137462_

Round 1

Reviewer 1 Report

This study addresses the overall impact on melanin formation processes of some lignin fractions, highlighting the multiple contribution fd one of these fraction capable of acting as antioxidant and inhibitor of the expression melanogenesis related proteins

in the opinion of this reviewer the study needs serious improvements at least in the presentation - here some more relevant issues:

1) the gel permeation chromatography was run on samples dissolved in THF but this include the insoluble acetone fraction - this produce a bias in the actually analyzed samples - and indeed results in terms of MW are very different from those of other studies

2) "In the results of the latter, the antioxi-233 dant activity of the acetone fraction was high, which is presumably because acetone has a higher polarity than methanol." --- methanol is much MORE POLAR than acetone

3)"For the tyrosinase activity’s inhibition by phenolic compounds, a possible mecha-262 nism of tyrosinase inhibition by lignin can be proposed because of its similar chemical 263 structure. According to the literature, lignin competitively reacts with tyrosinase to form 264 a lignin/oxytyrosinase complex, which inhibits the reaction of tyrosinase with L-DOPA 265 and then decreases the activity of the latter’s oxidation by tyrosinase. In this process, the 266 interaction between phenolic OH and copper is critical for the formation of the lignin/ox-267 ytyrosinase complex."

not at all clear - may be there is a superimposition of tyrosinase inhibition and inhibition of tyrosinase expression ?

4) "Kinetic analysis of lignin was conducted in a previous study, and 274 it was reported that both competitive and non-competitive inhibition were involved, as in 275 the results of this experiment [12]."

given the fractionation reported in this study are previous ones comparable with this ?

4.1) "dichloromethane fraction had the highest tyrosinase inhibition rate (75.12% in 0.5 mg/mL) 301 [11]. As described above,"

in particular in this study we do not have a dichloromethane fraction

5) "(obtained by alkali and ethanol extraction from three typical lig-337 nocellulosic materials) showed excellent tyrosinase inhibitory effects [13]. 338
339
340"

again, inhibition of tyrosinase or its expression ?

minor points:

1) "As shown in Figure 2 and Table 2, the SEL and lignin fractions 154 released similar peaks and diagnostic compounds" --- may be it should be "Figure 3"

2) "It is presumed that such a phenol structure exhibited high antioxidant 246 activity, and it was confirmed that the structure and activity of lignin differed depending 247 on the extraction and separation methods."

English here (but also elsewhere) needs revision -

Author Response

Answers to Reviewer’s Comments

We thank the reviewers for the feedback on our manuscript and for providing valuable comments that have benefitted the overall manuscript. We have revised the manuscript according to the comments made. Our responses (in red font) to each comment have been provided in the document that follows. Additionally, the revised sections of the manuscript are presented in red font, for your convenience.

Reviewer 1

This study addresses the overall impact on melanin formation processes of some lignin fractions, highlighting the multiple contribution for one of these fractions capable of acting as antioxidant and inhibitor of the expression melanogenesis related proteins.

in the opinion of this reviewer the study needs serious improvements at least in the presentation - here some more relevant issues:

q1) the gel permeation chromatography was run on samples dissolved in THF but this include the insoluble acetone fraction - this produce a bias in the actually analyzed samples - and indeed results in terms of MW are very different from those of other studies

a1) Thank you for the comment. THF is one of the solvents used in the GPC analysis of lignin. In this study, GPC analysis was performed on SEL-F1–F4 fractions, and the methanol-soluble SEL fraction (SEL-F2), which had the highest antioxidant and tyrosinase inhibitory activity, was used for the experiment. The molecular weight of the acetone-insoluble fraction (SEL-F4) was measured using THF as a solvent, as reported in a previous study (Ref: M.-F. Li, S.-N. Sun, F. Xu, R.-C. Sun, Sequential solvent fractionation of heterogeneous bamboo organosolv lignin for value-added application, Sep. Purif. Technol. 101 (2012) 18–25.). Acetylated lignin was measured, and the GPC system was used. In this case, the molecular weight was 11,820 g/mol, and no toxicity was observed.

q2) "In the results of the latter, the antioxidant activity of the acetone fraction was high, which is presumably because acetone has a higher polarity than methanol." --- methanol is much MORE POLAR than acetone

a2) The high antioxidant activity of the acetone-soluble SEL fraction (SEL-F3) was for DPPH radical scavenging ability. The antioxidant activity of methanol-soluble SEL fraction (SEL-F2) for the ABTS and hydroxyl radical scavenging activities was measured to be the highest.

q3)"For the tyrosinase activity’s inhibition by phenolic compounds, a possible mechanism of tyrosinase inhibition by lignin can be proposed because of its similar chemical structure. According to the literature, lignin competitively reacts with tyrosinase to form a lignin/oxytyrosinase complex, which inhibits the reaction of tyrosinase with L-DOPA and then decreases the activity of the latter’s oxidation by tyrosinase. In this process, the interaction between phenolic OH and copper is critical for the formation of the lignin/oxytyrosinase complex." not at all clear - may be there is a superimposition of tyrosinase inhibition and inhibition of tyrosinase expression ?

a3) The process of melanin synthesis proceeds in melanosomes in melanocytes. First, tyrosine is converted to DOPA quinone from DOPA by tyrosinase, and DOPA quinone is converted to DOPA chrome by tyrosinase to produce melanin. Thus, tyrosinase is a key enzyme involved in melanin synthesis. In addition, melanin is synthesized through intracellular signaling mechanisms, among which the cAMP/PKA pathway is the main pathway for melanin synthesis that also increases MITF expression via CREB. MITF, as an important transcriptional regulator, promotes the transcription of tyrosinase, TRP-1, and TRP-2 in the process of melanin synthesis. Therefore, inhibiting tyrosinase, an enzyme involved in melanin biosynthesis, is the most important key to whitening activity. It also inhibits the expression of transcription factors, such as tyrosinase, TRP-1, and TRP-2, involved in the upstream and downstream signaling processes of the melanin synthesis pathway.

q4) "Kinetic analysis of lignin was conducted in a previous study, and it was reported that both competitive and non-competitive inhibition were involved, as in the results of this experiment [12]." given the fractionation reported in this study are previous ones comparable with this ?

a4) In this experiment, lignin exhibited a graphical pattern of non-competitive inhibition. So, we edited the sentence as follows. " A previous study has conducted the kinetic analysis of lignin and reported that non-competitive inhibition was involved, similar to the results of this study [13]."

q4.1) "dichloromethane fraction had the highest tyrosinase inhibition rate (75.12% in 0.5 mg/mL) 301 [11]. As described above,"

in particular in this study we do not have a dichloromethane fraction

a4.1) Thank you for your comment. This has been cited from a previous study.

q5) "(obtained by alkali and ethanol extraction from three typical lig-337 nocellulosic materials) showed excellent tyrosinase inhibitory effects [13]. again, inhibition of tyrosinase or its expression ?

a5) Thank you for your comment. This has been cited from a previous study.

minor points:

q6) "As shown in Figure 2 and Table 2, the SEL and lignin fractions released similar peaks and diagnostic compounds" --- may be it should be "Figure 3"

a6) Thank you for your comment. It was a typographic error. We have revised it accordingly.

q7) "It is presumed that such a phenol structure exhibited high antioxidant activity, and it was confirmed that the structure and activity of lignin differed depending 247 on the extraction and separation methods." English here (but also elsewhere) needs revision -

A7) These phenolic structures are presumed to have high antioxidant activity, and the structure and activity of lignin were confirmed to be different depending on the extraction and separation methods. Since various phenolic structures are present in the structure of lignin, it shows high antioxidant activity, which varies depending on the extraction method and various structures of lignin.

Reviewer 2 Report

The manuscript represents an approach to counteract melanogenesis using lignin fractions. The introduction is well-written, and the authors followed a sound methodological approach to prove the research question. I just have some minor comments:

1- The readers could benefit from the citation of other approaches for treatment of melonogenesis in the introduction section. Some references of relevance can be

https://pubmed.ncbi.nlm.nih.gov/27356545/

https://www.ncbi.nlm.nih.gov/pmc/articles/PMC7271691/

https://www.sciencedirect.com/science/article/abs/pii/S1773224720312739

https://www.mdpi.com/2079-9284/8/3/82/htm

2- It is not clear in Table 1 whether the values represent mean and S.D. or mean and S.E. Please clarify.

3- Figure 5 looks blurry and needs to be uploaded in better resolution.

4- How does the safety of lignin fraction 2 compare to that of arbutin? if there are reports stating that arbutin is not safe on cells, then this can be a point of advantage for the lignin fraction and would be worth mentioning. 

5- The manuscript needs to be revised by a native speaker for correction of some grammatical errors.

Author Response

Answers to Reviewer’s Comments

We thank the reviewers for the feedback on our manuscript and for providing valuable comments that have benefitted the overall manuscript. We have revised the manuscript according to the comments made. Our responses (in red font) to each comment have been provided in the document that follows. Additionally, the revised sections of the manuscript are presented in red font, for your convenience.

Reviewer 2

Q1) The manuscript represents an approach to counteract melanogenesis using lignin fractions. The introduction is well-written, and the authors followed a sound methodological approach to prove the research question. I just have some minor comments:

1- The readers could benefit from the citation of other approaches for treatment of melonogenesis in the introduction section. Some references of relevance can be

https://pubmed.ncbi.nlm.nih.gov/27356545/

https://www.ncbi.nlm.nih.gov/pmc/articles/PMC7271691/

https://www.sciencedirect.com/science/article/abs/pii/S1773224720312739

https://www.mdpi.com/2079-9284/8/3/82/htm

A1) Thank you for the comment. We included the citation in the manuscript.

Q2) It is not clear in Table 1 whether the values represent mean and S.D. or mean and S.E. Please clarify.

A2) Table 1. showed the standard deviation (S.D).

Q3) Figure 5 looks blurry and needs to be uploaded in better resolution.

A3) We changed the figure into better one.

Q4) How does the safety of lignin fraction 2 compare to that of arbutin? if there are reports stating that arbutin is not safe on cells, then this can be a point of advantage for the lignin fraction and would be worth mentioning. 

A4) We added discussion on that in the manuscript.

Q5) The manuscript needs to be revised by a native speaker for correction of some grammatical errors.

A5) We have got English editing service by a native speaker. And we changed the title according to the comment.

Reviewer 3 Report

Introduction

Hydroquinone, kojic acid and arbutin: are not all the same

Hydroquinone is cytotoxic and banned for use in cosmetics. Kojic acid is unstable (kojic acid dipalmitate is widely used) and arbutin is under consideration. Both Kojic acid and arbutin remain cosmetic ingredients

Although there is an extented report about bamboos  in the introduction, there is a lack of arguments that supprt the need of the discovery of new skin-lighteners  cosmetic ingredients 

Conclusions-discussion

In general, I feel more discussion is needed for the correlation between   the different antioxidant results in the different methods used and the inhibition of tyrosinase activity.

Author Response

Answers to Reviewer’s Comments

We thank the reviewers for the feedback on our manuscript and for providing valuable comments that have benefitted the overall manuscript. We have revised the manuscript according to the comments made. Our responses (in red font) to each comment have been provided in the document that follows. Additionally, the revised sections of the manuscript are presented in red font, for your convenience.

Reviewer 3

Introduction

Q1) Hydroquinone, kojic acid and arbutin: are not all the same

Hydroquinone is cytotoxic and banned for use in cosmetics. Kojic acid is unstable (kojic acid dipalmitate is widely used) and arbutin is under consideration. Both Kojic acid and arbutin remain cosmetic ingredients

A1) Thank you for the comment. Hydroquinone, a tyrosinase inhibitor also used as a cosmetic whitening agent, is currently banned as a cosmetic ingredient as it is a potential carcinogen. In addition, Kojic acid, which inhibits tyrosinase and is a representative whitening substance, is a potential allergen. Compared to other whitening functional ingredients, arbutin causes less skin irritation and is a highly safe ingredient, but it may have minor side effects depending on individual skin characteristics.

Conclusions-discussion

Q2) In general, I feel more discussion is needed for the correlation between   the different antioxidant results in the different methods used and the inhibition of tyrosinase activity.

A2) Active oxygen, such as ROS, causes photoaging of the skin and produces age spots, spots, and freckles. Moreover, the elimination of active oxygen correlates with the anti-whitening effect. Natural extracts have various effects such as UV absorption, anti-inflammatory action, antioxidant action, inhibition of enzymes such as collagenase and tyrosinase, and percutaneous absorption improvement [46]. Recently, side effects such as the production of phenolic synthetic antioxidants BHA (butylated hydroxyanisole) and BHT (butylated hydroxytoluene) have been revealed, and many studies have focused on developing antioxidants from natural sources (M.H. Choi, H.J. Shin, Anti-melanogenesis effect of quercetin, Cosmetics 2016 3 18–34.).

Round 2

Reviewer 1 Report

the general view of the ms. appears not changed, but authors addressed the raised issues and actually the study can be published